# COUNTERFACTUAL FAIRNESS WITH HUMAN IN THE LOOP

## ABSTRACT

Machine learning models have been increasingly used in human-related applications such as healthcare, lending, and college admissions. As a result, there are growing concerns about potential biases against certain demographic groups. To address the unfairness issue, various fairness notions have been introduced in the literature to measure and mitigate such biases. Among them, Counterfactual Fairness (CF) Kusner et al. (2017) is a notion defined based on an underlying causal graph that requires the prediction perceived by an individual in the real world to remain the same as it would be in a counterfactual world, in which the individual belongs to a different demographic group. Unlike Kusner et al. (2017), this work studies the long-term impact of machine learning decisions using a causal inference framework where the individuals' future status may change based on the current predictions. We observe that imposing the original counterfactual fairness may not lead to a fair future outcome for the individuals. We thus introduce a fairness notion called *lookahead counterfactual fairness* (LCF), which accounts for the downstream effects of ML models and requires the individual *future status* to be counterfactually fair. We theoretically identify conditions under which LCF can be improved and propose an algorithm based on our theoretical results. Experiments on both synthetic and real data show the effectiveness of our method.

## 1 INTRODUCTION

The integration of machine learning (ML) into high-stakes domains (e.g., loan lending, hiring, college admissions, healthcare) has the potential to enhance traditional human-driven processes. However, it may introduce biases and treat protected groups unfairly. For instance, it has been shown that the violence risk assessment tool, SAVRY, discriminates against males and foreigners (Tolan et al., 2019); the previous Amazon hiring system exhibits gender bias (Dastin, 2018); the accuracy of a computer-aided clinical diagnostic system highly depends on the race of patients (Daneshjou et al., 2021). To address unfairness issues, numerous fairness notions have been proposed, including *unawareness* that prevents the explicit use of demographic attributes, *parity-based fairness* that requires certain statistics (e.g., accuracy, true/false positive rate) to be equal across different groups (Hardt et al., 2016b), *preference-based fairness* that ensures individuals would collectively prefer their perceived outcomes regardless of the (dis)parity compared to other groups (Zafar et al., 2017; Do et al., 2022). However, these notions often overlook the underlying causal structures among different variables. In contrast, Kusner et al. (2017) introduced the concept of *counterfactual fairness* (CF), which posits that an individual should receive consistent treatment in a counterfactual world where their sensitive attribute differs. Chiappa (2019), Zuo et al. (2022), Wu et al. (2019), Xu et al. (2019) and Ma et al. (2023) are also among recent efforts to take into account the causal structure while training a fair predictor.

Yet, CF is primarily studied in static settings without considering the consequences of machine learning decisions. In several applications, ML decisions can change future data distribution. For example, Ensign et al. (2018) shows that the use of predictive policing systems for allocating law enforcement resources increases the likelihood of uncovering crimes in regions with a greater concentration of policing resources. When we design an ML system, we should take into account that such a system interacts with individuals, and individuals may subsequently adapt their behaviors and modify the features in response to the ML system (Miller et al., 2020; Shavit et al., 2020). As a result, learning (fair) models in a static setting without accounting for such downstream ef-

fects may lead to unexpected adverse consequences. Although there are several works that consider the long-term impact of fair decisions Henzinger et al. (2023a); Ge et al. (2021); Henzinger et al. (2023b), the long-term impact of counterfactually fair decisions has not been studied extensively. The most related work to this work is (Hu and Zhang, 2022) which uses path-specific effects as a measure of fairness in a sequential framework where individuals change features while interacting with an ML system over time, and the goal is to ensure ML *decisions* satisfy the fairness constraint throughout the sequential process. However, Hu and Zhang (2022) do not argue how the data distribution changes over time and whether this change decreases the disparity in the long run. For example, Zhang et al. (2018) show that the statistical parity fairness notion may worsen the disparity in a sequential setting if it is not aligned with the factors that derive user dynamics. As a result, in this work, we are interested in understanding how individuals are affected by an ML system using a counterfactual inference framework. We argue that imposing counterfactually fair ML decisions may not necessarily decrease the disparity in a sequential setting when we compare factual and counterfactual worlds.

In this work, we focus on fairness evaluated over individual *future* status (label), which accounts for the downstream effects of ML decisions on individuals. We aim to examine under what conditions and by what algorithms the disparity between individual future status in factual and counterfactual worlds can be mitigated after deploying ML decisions. To this end, we first introduce a new fairness notion called "lookahead counterfactual fairness (LCF)." Unlike the original counterfactual fairness proposed by Kusner et al. (2017) that requires the ML predictions received by individuals to be the same as those in the counterfactual world, LCF takes one step further by enforcing the individual future status (after responding to ML predictions) to be the same.

Given the definition of LCF, we then develop algorithms that learn ML models under LCF. To model the effects of ML decisions on individuals, we focus on scenarios with strategic individuals who respond to ML models by increasing their chances of receiving favorable decisions; this can be mathematically represented by modifying their features toward the direction of the gradient of the decision function (Rosenfeld et al., 2020). We first theoretically identify conditions under which an ML model can satisfy LCF, and then develop an algorithm for training ML models under LCF.

Our contributions can summarized as follows:

- We propose a novel fairness notion that focuses on the counterfactual fairness over individual future status (i.e., actual labels after responding to ML systems). Unlike the previous counterfactual fairness notion that focuses on ML decisions, this notion accounts for the subsequent impacts of ML decisions and aims to ensure fairness over individual actual future status.

- For scenarios where individuals respond to ML models by adjusting their features toward the direction of the gradient of decision functions, we theoretically identify conditions under which an ML model can satisfy LCF. We further develop an algorithm for training an ML model under LCF.

- We conduct extensive experiments on both synthetic and real data to validate the proposed algorithm. Results show that compared to conventional counterfactual fair predictors, our method can improve disparity with respect to the individual actual future status.

## 2 PROBLEM FORMULATION

We consider a supervised learning problem with a training dataset consisting of triples $(A, X, Y)$, where $A \in \mathcal{A}$ is a sensitive attribute (e.g., race, gender), $X = [X_1, X_2, ..., X_d]^{\mathrm{T}} \in \mathcal{X}$ is a $d$-dimensional feature vector, and $Y \in \mathcal{Y} \subseteq \mathbb{R}$ is the target variable indicating individual's underlying status (e.g., $Y$ in lending identifies the applicants' abilities to repay the loan, $Y$ in healthcare may represent patients' insulin spike level). The goal is to learn a predictor from training data that can predict $Y$ given inputs $A$ and $X$. Let $\hat{Y}$ denote as the output of the predictor. We further assume that $(A, X, Y)$ is associated with a structural causal model (SCM) (Pearl et al., 2000) $\mathcal{M} = (V, U, F)$, where $V = (A, X, Y)$ represents observable variables, $U$ includes unobservable (exogenous) variables that are not caused by any variable in $V$, and $F = \{f_1, f_2, \ldots, f_{d+2}\}$ is a set of $d+2$ functions called *structural equations* that determines how each observable variable is constructed. More precisely, we have the following structural equations,

$$X_i = f_i(pa_i, U_{pa_i}), \ \forall i \in \{1, \cdots, d\}, \ A = f_A(pa_A, U_{pa_A}), \ Y = f_Y(pa_Y, U_{pa_Y}), \quad (1)$$

where $pa_i \subseteq V$, $pa_A \subseteq V$ and $pa_Y \subseteq V$ are observable variables that are the parents of $X_i$, $A$, and $Y$, respectively. $U_{pa_i} \subseteq U$ are unobservable variables that are the parents of $X_i$. Similarly, we denote unobservable variables $U_A \subseteq U$ and $U_Y \subseteq U$ as the parents of $A$ and $Y$, respectively.

## 2.1 BACKGROUND: COUNTERFACTUALS

If the probability density functions of unobserved variables are known, we can leverage the structural equations in SCM to find the marginal distribution of any observed variable $V_i \in V$ and even study how intervening certain observed variables impacts other variables. Specifically, the **intervention** on variable $V_i$ is equivalent to replacing structural equation $V_i = f_i(pa_i, U_{pa_i})$ with equation $V_i = v$ for some $v$. Given new structural equation $V_i = v$ and other unchanged structural equations, we can find out how the distribution of other observable variables changes as we change value $v$.

In addition to understanding the impact of an intervention, SCM can further facilitate **counterfactual inference**, which aims to answer the question "*what would be the value of $Y$ if $Z$ had taken value $z$ in the presence of evidence $O = o$ (both $Y$ and $Z$ are two observable variables)?*" Specifically, given $U = u$ and structural equations $F$, the counterfactual value of $Y$ can be computed by replacing the structural equation for $Z$ as $Z = z$ and replacing $U$ with $u$ in the rest of the structural equations. Such counterfactual value is typically denoted by $Y_{Z \leftarrow z}(u)$. Given evidence $O = o$, the distribution of counterfactual value $Y_{Z \leftarrow z}(U)$ can be calculated as follows,[1]

$$\Pr\{Y_{Z \leftarrow z}(U) = y | O = o\} = \sum_u \Pr\{Y_{Z \leftarrow z}(u) = y\} \Pr\{U = u | O = o\} \tag{2}$$

**Example 2.1** (**Law School Success**). Consider two demographic groups of college students distinguished by gender whose first-year average (FYA) in college is denoted by $Y$. The FYA of each student is causally related to (observable) grade-point average (GPA) before entering college, entrance exam score (LSAT), and gender. Denote gender by $A \in \{0, 1\}$, GPA by $X_G$, and LSAT by $X_L$. Suppose there are two unobservable variables $U = (U_A, U_{XY})$, e.g., $U_{XY}$ may be interpreted as the student's knowledge. Consider the following structural equations:

$$
\begin{aligned}
A &= U_A, & X_G &= b_G + w_G^A A + U_{XY}, \\
X_L &= b_L + w_L^A A + U_{XY}, & Y &= b_F + w_F^A A + U_{XY},
\end{aligned}
$$

where $(b_G, w_G^A, b_L, w_L^A, b_F, w_F^A)$ are know parameters of the causal model. Given observation $X_G = 1, A = 0$, the counterfactual value can be calculated with an abduction-action-prediction procedure. It is easy to see that $U_{XY} = 1 - b_G$ and $U_A = 0$ with probability 1.0 (Abduction). Substitute $A$ with 1 (Action). As a result, the counterfactual value of $Y_{A \leftarrow 1}(U)$ given $X_G = 1, A = 0$ can be calculated as follows (Action),

$$Y_{A \leftarrow 1}(U) = b_f + w_F^A + 1 - b_G \quad \text{with probability 1.0}$$

## 2.2 COUNTERFACTUAL FAIRNESS

Counterfactual Fairness (CF) has been proposed by Kusner et al. (2017) which requires that for an individual with $(X = x, A = a)$, the prediction $\hat{Y}$ in the factual world should be the same as that in the counterfactual world in which the individual belongs to a demographic group other than $A = a$. Mathematically, the counterfactual fairness is defined as follows: $\forall a, \check{a} \in \mathcal{A}, X \in \mathcal{X}, y \in \mathcal{Y}$,

$$\Pr\left(\hat{Y}_{A \leftarrow a}(U) = y | X = x, A = a\right) = \Pr\left(\hat{Y}_{A \leftarrow \check{a}}(U) = y | X = x, A = a\right),$$

While the CF notion has been widely used in the literature, it does not take into account the downstream impacts of ML prediction $\hat{Y}$ on individuals in factual and counterfactual worlds. To illustrate the importance of considering such impacts, we provide an example.

**Example 2.2.** Consider a loan approval problem where an applicant with $(X = x, A = a)$ applies for the loan and the goal is to predict the applicant's ability to repay the loan. As highlighted by Liu

---

[1]Note that given structural equations (equation 1) and marginal distribution of $U$, $\Pr\{U = u, O = o\}$ can be calculated using the Change-of-Variables Technique and the Jacobian factor. As a result, $\Pr\{U = u | O = o\} = \frac{\Pr\{U=u, O=o\}}{\Pr\{O=o\}}$ can be also calculated accordingly.

et al. (2018), issuing loans to unqualified people who cannot repay the loan may hurt them by worsening their future credit scores. Assume that this person in the factual world is qualified for the loan and does not default. However, in the counterfactual world where the individual belongs to another demographic group, he/she is not qualified. Under counterfactually fair predictions, both individuals in the factual and counterfactual world should receive the same distribution of decision. If both individuals are issued a loan, the one in the counterfactual world would suffer from a worse credit score in the future. Therefore, it is essential to account for the downstream effects of predictions when learning a fair ML model.

Motivated by the above example, this work aims to study CF in a dynamic setting where the deployed ML decisions may affect individual behavior and change their future features and underlying statuses. Formally, we assume individuals after receiving prediction $\hat{Y}$, their future feature vector $X'$ is determined by a response function $r : \mathcal{X} \times \mathcal{Y} \to \mathcal{X}$,

$$X_i' = r(X_i, \hat{Y}), \tag{3}$$

We assume the structural equation $f_Y$ for the target variable (underlying status) $Y$ remains fixed, so that individuals' feature changes also cause their statuses to change. Denote $Y'$ as the consequent future status generated by $f_Y$ and new features $X'$. One way to tackle the issue in Example 2.2 is to explicitly consider the individual response and impose the fairness constraint on future status $Y'$ instead of the prediction $\hat{Y}$. We call such a fairness notion *Lookahead Counterfactual Fairness (LCF)* and present it in the next section.

## 2.3 LOOKAHEAD COUNTERFACTUAL FAIRNESS

To take into account the downstream impacts of ML decision, we impose the fairness constraint on future outcome $Y'$. Given structural causal model $\mathcal{M} = (U, V, F)$, individual response function $r$, and data $(A, X, Y)$, we define lookahead counterfactual fairness below.

**Definition 2.1.** We say an ML model satisfies lookahead counterfactual fairness (LCF) if $\forall a, \breve{a} \in \mathcal{A}, X \in \mathcal{X}, y \in \mathcal{Y}$, the following holds:

$$\Pr(Y'_{A \leftarrow a}(U) = y | X = x, A = a) = \Pr(Y'_{A \leftarrow \breve{a}}(U) = y | X = x, A = a), \tag{4}$$

LCF implies that the *downstream consequence* of ML decisions for a given individual in the factual world should be the same as that in the counterfactual world where the individual belongs to other demographic groups. Note that CF may contradict LCF: even under counterfactually fair predictor, individuals in the factual and counterfactual worlds may end up with very different future statuses. We show this with an example below.

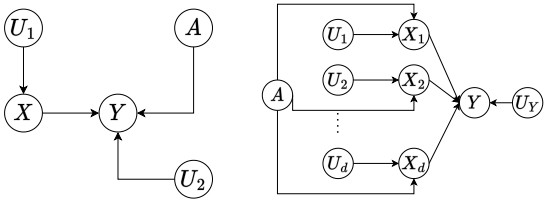

(a) Causal Graph - Type 1      (b) Causal Graph - Type 2

Figure 1: Two type of Causal Graphs

**Example 2.3.** Consider the causal graph in Figure 1a. Based on Kusner et al. (2017), a predictor that only uses $U_1$ and $U_2$ as input is counterfactually fair.[2] Therefore, $\hat{Y} = h(U_1, U_2)$ is a counterfactually fair predictor. Suppose the structural functions are as follows,

$$X = f_X(U_1) = U_1, \qquad\qquad Y = f_Y(U_2, X, A) = U_2 + X + A$$
$$U_1' = r(U_1, \hat{Y}) = U_1 + \nabla_{U_1}\hat{Y}, \qquad U_2' = r(U_2, \hat{Y}) = U_2 + \nabla_{U_2}\hat{Y}$$
$$X' = f_X(U_1') = U_1', \qquad\qquad Y' = f_Y(U_2', X', A) = U_2' + X' + A$$

The prior distributions of $U_1$ and $U_2$ are the uniform distributions over $[-1, 1]$. Note that the response functions stated above imply that individuals make efforts toward changing feature vectors through changing the unobservable variables, which results in higher $\hat{Y}$ in the future. It is easy to see that

---

[2]Note that $U_1$ and $U_2$ can be generated for each sample $(X, A)$. Please see Section 4.1 of Kusner et al. (2017) for more details.

$h(U_1, U_2) = U_1 + U_2$ minimizes the MSE loss $\mathbb{E}\{(Y - \hat{Y})^2\}$ if $A \in \{-1, 1\}$ and $\Pr\{A = 1\} = 0.5$. However, since $\nabla_{U_1}\hat{Y} = \nabla_{U_2}\hat{Y} = 1$, we have the following equations,

$$
\begin{aligned}
\Pr(Y'_{A \leftarrow a}(U) = y | X = x, A = a) &= \delta(y - a - x - 2) \\
\Pr(Y'_{A \leftarrow \check{a}}(U) = y | X = x, A = a) &= \delta(y - \check{a} - x - 2)
\end{aligned}
$$

where $\delta(y) = \begin{cases} 1 & \text{if } y = 0 \\ 0 & \text{o.w.} \end{cases}$. It shows that although the decisions in the factual and counterfactual worlds are the same, the future status $Y'$ are still different and Definition 2.1 does not hold One of the general case where CF holds but LCF not holds is shown in Appendix A.5.

## 3  LEARNING UNDER LCF

In this work, we focus on a class of response functions $r$ that an individual responds to ML prediction by increasing the prediction made by the ML model. Note that such type of response has also been widely studied in strategic classification, e.g., Rosenfeld et al. (2020); Hardt et al. (2016a). In particular, we consider the response function in the following form,

$$
\begin{aligned}
U'_i &= r(U_i, \hat{Y}) = U_i + \eta \nabla_{U_i}\hat{Y}, \quad \forall U_i \in U \\
X'_i &= r(X_i, \hat{Y}) = X_i + \eta \nabla_{X_i}\hat{Y}, \quad X_i \text{ is a root node.}
\end{aligned}
$$

Here, we are assuming that root node variables in the causal graph are updated based on response function $r$ and decision variable $\hat{Y}$. The effects of $\hat{Y}$ on other variables are passed through the causal structural equations. Our goal is to train a model under LCF constraint. Before presenting our algorithm, we first define the notion of counterfactual random variables.

**Definition 3.1** (Counterfactual Random Variables). Let $x$ and $a$ be the realization of random variables $X$ and $A$, and $\check{a} \neq a$. We say $\check{X} := X_{A \leftarrow \check{a}}(U)$ and $\check{Y} := Y_{A \leftarrow \check{a}}(U)$ are the counterfactual random variables associated with $(x, a)$ if $U$ follows the conditional distribution $\Pr\{U | X = a, A = a\}$ as given by the causal Model $\mathcal{M}$. The realization of $\check{X}$ and $\check{Y}$ are denoted by $\check{x}$ and $\check{y}$.

The following theorem constructs a predictor $g$ to satisfy LCF. That is, under predictor $g$ proposed in Theorem 3.1, $Y'$ satisfies Definition 2.1.

**Theorem 3.1.** Consider a structural causal model $\mathcal{M} = (U, V, F)$, where $U = \{U_X, U_Y\}$, $U_X = [U_1, U_2, ..., U_d]^{\mathrm{T}}$, $V = \{A, X, Y\}$, $X = [X_1, X_2, ..., X_d]^{\mathrm{T}}$. Assume that the structural functions are given by (see Figure.1b, where exogenous variables do not include unobserved confounders),

$$
X = \alpha \odot U_X + \beta A, \quad Y = w^{\mathrm{T}} X + \gamma U_Y, \tag{5}
$$

where $\alpha = [\alpha_1, \alpha_2, ..., \alpha_d]^{\mathrm{T}}$, $\beta = [\beta_1, \beta_2, ..., \beta_d]^{\mathrm{T}}$, $w = [w_1, w_2, .., w_d]^{\mathrm{T}}$, and $\odot$ denotes the element wise production. Then, the following predictor satisfies LCF defined in Definition 2.1,

$$
g(\check{Y}, U) = p_1 \check{Y}^2 + p_2 \check{Y} + p_3 + h(U), \tag{6}
$$

where $p_1 = \frac{T}{2}$ with $T := \frac{1}{\eta(||w \odot \alpha||^2 + \gamma^2)}$, and $h$ is an arbitrary function (e.g., a neural network).

It is worth mentioning that Definition 2.1 can be a very strong constraint in scenarios when $Y_{A \leftarrow a}(U)$ and $Y_{A \leftarrow \check{a}}$ have significantly different distributions. In this case, enforcing $Y'_{A \leftarrow a}(U)$ and $Y'_{A \leftarrow \check{a}}(U)$ to have the same distributions may degrade the performance of the predictor significantly. As a result, we can also consider a weaker version of Definition 2.1 stated below.

**Definition 3.2.** We say lookahead counterfactual fairness improves if the following holds,

$$
\Pr\left( \left\{ |Y'_{A \leftarrow a}(U) - Y'_{A \leftarrow \check{a}}(U)| < |Y_{A \leftarrow a}(U) - Y_{A \leftarrow \check{a}}(U)| \right\} | X = x, A = a \right) = 1, \tag{7}
$$

$$
\forall (a, \check{a}) \in \mathcal{A}^2, a \neq \check{a}, X \in \mathcal{X}, y \in \mathcal{Y}.
$$

Definition 3.2 implies that after individual response, the difference between future status $Y'$ in factual and counterfactual worlds should be smaller than the difference between original status $Y$ in factual and counterfactual worlds. Generally speaking, this implies that the disparity between factual and counterfactual worlds must get better over time. As we show in our experiments, constraint 7 is weaker than constraint 4 and can lead to better prediction performance.

**Theorem 3.2.** Consider the structural causal model with structural equations described in equation 5. A predictor $g(\check{Y}, U)$ improves lookahead counterfactual fairness (i.e., future status $Y'$ satisfies constraint 7) if $g$ has the following three properties:

- $g(\check{y}, u)$ is strictly convex w.r.t. $\check{y}$.

- $g(\check{y}, u)$ can be expressed as: $g(\check{y}, u) = g_1(\check{y}) + g_2(u)$.

- The derivative of $g(\check{y}, u)$ w.r.t. $\check{y}$ is $K$-Lipschitz continuous in $\check{y}$ with $K < \frac{2}{\eta(||w \odot \alpha||^2 + \gamma^2)}$, $|\frac{\partial g(\check{y}_1, u)}{\partial \check{y}} - \frac{\partial g(\check{y}_2, u)}{\partial \check{y}}| \leq K|\check{y}_1 - \check{y}_2|$.

Theorems 3.1 and 3.2 shed light on how to train a predictor under constraints 4 and 7. Specifically, given a training dataset $D = \{(x^{(i)}, y^{(i)}, a^{(i)})\}_{i=1}^n$, we first estimate the structural equations. Then, we choose a parameterized predictor $g$ that satisfies the conditions in Theorem 3.1 or Theorem 3.2. An example is shown in Algorithm 1, which finds an optimal predictor in the form of $g(\check{y}, u) = p_1 \check{y}^2 + p_2 \check{y} + p_3 + h_\theta(u)$, where $p_1$ is a hyperparameter, $\theta$ is the training parameter for function $h$, and $p_2, p_3$ are two other training parameters. Under Algorithm 1, we can find the optimal values for $p_2, p_3, \theta$ using training data $D$. Note that since $p_1$ is a hyperparameter, we can control the strength of fairness by choosing its value, e.g., to satisfy Definition 2.1, we should set $p_1 = T/2$ based on Theorem 3.1. If we want to satisfy Definition 3.2, we should choose $0 < p_1 < T$ to make sure $g$ satisfies the first and third conditions in Theorem 3.2.

---

**Algorithm 1** Training a Predictor under LCF

---

**Input:** Training dataset $D = \{(x^{(i)}, y^{(i)}, a^{(i)})\}_{i=1}^n$.
 1: Estimate the structural equations 5 using $D = \{(x^{(i)}, y^{(i)}, a^{(i)})\}_{i=1}^n$ to determine parameters $\alpha$, $\beta$, $w$, and $\gamma$.
 2: For each data point $(x^{(i)}, y^{(i)}, a^{(i)})$, draw $m$ samples $u^{(i)[j]}, j = 1, \ldots, m$ from conditional distribution $U|X = x^{(i)}, A = a^{(i)}$ and generate counterfactual $\check{y}^{(i)[j]}$ corresponding to $u^{(i)[j]}$ based on structural equations 5.
 3: Solve the following optimization problem,

$$\hat{p}_2, \hat{p}_3, \hat{\theta} = \arg\min_{p_2, p_3, \theta} \frac{1}{mn} \sum_{i=1}^n \sum_{j=1}^m l(g(\check{y}^{(i)[j]}, u^{(i)[j]}), y^{(i)}), \tag{8}$$

where $g(\check{y}^{(i)[j]}, u^{(i)[j]}) = p_1(\check{y}^{(i)[j]})^2 + p_2 \check{y}^{(i)[j]} + p_3 + h_\theta(u)$, $\theta$ is a parameter for function $h$, and $l$ is a loss function.
**Output:** $\hat{p}_2, \hat{p}_3, \hat{\theta}$

---

Note that the results we have so far are for the linear causal models. When the causal model is non-linear, it is hard to construct a model satisfying *perfect* LCF (Definition 2.1). Nonetheless, we can still show that it is possible to *improve* LCF (Definition 3.2) for certain non-linear causal models.

**Theorem 3.3** (Informal). Consider a causal model $\mathcal{M}$ with the following assumed structural equations,

$$X_i = A(\alpha_i U_i + \beta_i), \ i \in \{1, 2, ..., d\}, \quad Y = w^T X + \gamma U_Y$$

Let $\check{Y}$ be the counterfactual output associated with $X = x, A = a$. Consider predictor $g(\check{Y})$ which is a strictly convex and twice differentiable function and only uses the counterfactual random variable $\check{Y}$ as input. Then, under certain conditions on the derivatives of $g$ and the properties of $A$, predictor $g(\check{Y})$ improves LCF (i.e., a future status $Y'$ satisfies Definition 3.2).

Due to the page limit, the specific conditions of Theorem 3.3 are formally stated in Appendix A.3. And we also show another the possibility of improving LCF for another kind of causal model in Appendix A.6.

## 4 EXPERIMENTS

We conduct experiments on both synthetic and real data to validate the proposed method.

### 4.1 SYNTHETIC DATA

In this section, we use synthetic data to test our proposed method. The synthetic data has been generated based on the causal model described in Theorem 3.1. We set $d$ equal to 10 and generated 1000 data points. We assume that $U_X$ and $U_Y$ follow the uniform distribution over $[0, 1]$. Also, sensitive attribute $A \in \{0, 1\}$ is a Bernoulli random variable with $\Pr\{A = 0\} = 0.5$. Then, we generate $X$ and $Y$ using the structural functions described in equation 5.[3] Based on the causal model, the conditional distribution of $U_X$ and $U_Y$ given $X = x, A = a$ are as follows,

$$U_X | X = a, A = a \sim \delta(\frac{x - \beta a}{\alpha}) \qquad U_Y | X = x, A = a \sim \text{Uniform}(0, 1) \qquad (9)$$

**Baselines**: We used two baselines for comparison. The first one is the Unfair predictor (UF), which is a linear model and ignores the fairness constraint. The UF model gets feature $X$ as input and predicts $Y$. The second one is the counterfactual fair predictor (CF), which takes only the unobservable variables $U$ as the input. This predictor has been introduced by Kusner et al. (2017).

**Implementation Details**: To find a predictor satisfying Definition 2.1, we train a predictor in the form of equation 6. In our experiment, $h(u)$ is a linear function. To train $g(\check{y}, u)$, we follows Algorithm 1 with $m = 100$. We split the dataset into the training/validation/test set at $60\%/20\%/20\%$ ratio randomly and repeat the experiment 5 times. We use the validation set to find the optimal number of training epochs and the learning rate. Based on our observation, Adam optimization with a learning rate equal to $10^{-3}$ and 2000 epochs gives us the best performance.

**Metrics**: We use three metrics to evaluate the methods. To evaluate the performance, we use the mean squared error (MSE). Given a dataset $\{x^{(i)}, a^{(i)}, y^{(i)}\}_{i=1}^n$, for each $x^{(i)}$ and $a^{(i)}$, we generate $m = 100$ values of $u^{(i)[j]}$ from the posterior distribution. MSE can be estimated as follows,[4]

$$\text{MSE} = \frac{1}{mn} \sum_{i=1}^n \sum_{j=1}^m ||y^{(i)} - \hat{y}^{(i)[j]}||^2, \qquad (10)$$

where $\hat{y}^{(i)[j]}$ is the prediction for datapoint data $(x^{(i)}, a^{(i)}, u^{(i)[j]})$. Note that for the UF baseline, the prediction does not depend on $u^{(i)[j]}$. Therefore, $\hat{y}^{(i)[j]}$ does not change by $j$ for the UF predictor. To evaluate fairness, we define a metric called average future causal effect (AFCE),

$$\text{AFCE} = \frac{1}{mn} \sum_{i=1}^n \sum_{j=1}^m |y'^{(i)[j]} - \check{y}'^{(i)[j]}| \qquad (11)$$

It is the average difference between the factual and counterfactual future outcomes. In order to compare $|Y - \check{Y}|$ with $|Y' - \check{Y}'|$ under different algorithms, we present the following metric called unfairness improvement ratio (UIR),

$$\text{UIR} = (1 - \frac{\sum_{i=1}^n \sum_{j=1}^m |y'^{(i)[j]} - \check{y}'^{(i)[j]}|}{\sum_{i=1}^n \sum_{j=1}^m |y^{(i)[j]} - \check{y}^{(i)[j]}|}) \times 100\%. \qquad (12)$$

Larger UIR implies a higher improvement in disparity.

**Results**: Table1 illustrates the results when we set $\eta = 10$ and $p_1 = \frac{T}{2}$. The results show that our method can achieve perfect LCF with $p_1 = \frac{T}{2}$. Note that in our experiment, the range of $Y$ is $[0, 3.73]$, and our method and UF can achieve similar MSE. Note that our method achieves better performance compared to the CF method because $\check{Y}$ includes useful predictive information and using it in our predictor can improve performance and decrease the disparity at the same time. Both CF and UF do not take into account future outcome $Y'$, and as a result, $|Y' - \check{Y}'|$ is similar to $|Y - \check{Y}|$ leading UIR $= 0$. Based on what we discussed in the proof of

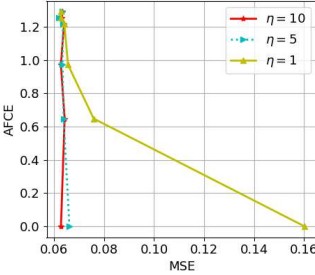

Figure 2: MSE-AFCE Trade-off

---

[3]The exact values for parameters $\alpha$, $\beta$, $w$ and $\gamma$ can be found in the appendix.
[4]Check Section 4.1 of Kusner et al. (2017) for details on why equation 10 is an empirical estimate of MSE.

Table 1: Results on Synthetic Data: comparison with two baselines, unfair predictor (UF) and counterfactual fair predictor (CF), in terms of performance (MSE) and LCF (AFCE ,UIR).

| Method | MSE | AFCE | UIR |
|--------|-----|------|-----|
| UF | $0.036 \pm 0.003$ | $1.296 \pm 0.000$ | $0\% \pm 0$ |
| CF | $0.520 \pm 0.045$ | $1.296 \pm 0.000$ | $0\% \pm 0$ |
| Ours ($p_1 = T/2$) | $0.064 \pm 0.001$ | $0.000 \pm 0.0016$ | $100\% \pm 0$ |

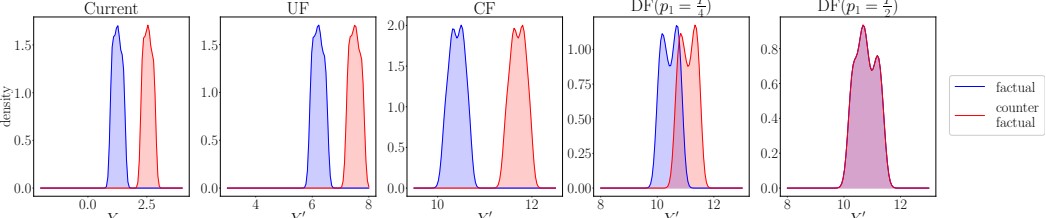

Figure 3: Density plot for $Y'$ and $\check{Y}'$ under the distribution of $U$ in synthetic data

Theorem 3.1, the value of $p_1$ can impact the accuracy-fairness trade-off. We changed the value of $p_1$ from $\frac{T}{2}$ to $\frac{T}{512}$ for different values of $\eta$ and calculated MSE as a function of AFCE in Figure 2. It shows that we can easily control accuracy-fairness trade-off in our algorithm by changing $p_1$ from $T/2$ to 0. To show how our method impacts a specific individual, we choose the first data point in our test dataset and plot the distribution of factual future status $Y'$ and counterfactual future status $\check{Y}'$ for this specific data point under different methods. Figure 3 illustrates such distributions. It can be seen in the most left plot that there is an obvious gap between factual $Y$ and counterfactual $\check{Y}$. Both UF and CF can not decrease this gap for future outcome $Y'$. However, with our method, we can observe that the distributions of $Y'$ and $\check{Y}'$ become closer to each other. When $p_1 = \frac{T}{2}$ (the most right plot in Figure 3), the two distributions become the same in the factual and counterfactual worlds.

## 4.2 REAL DATA: THE LAW SCHOOL SUCCESS DATASET

We further measure the performance of our proposed method using the Law School Admission Dataset Wightman (1998). In this experiment, the objective is to forecast the first-year average grades (FYA) of students in law school using their undergraduate GPA and LSAT scores.

**Dataset**: The dataset consists of 21,791 records. Each record is characterized by 4 attributes: Sex ($S$), Race ($R$), UGPA ($G$), LSAT ($L$), and FYA ($F$). Both Sex and Race are categorical in nature. The Sex attribute can be either male or female, while Race can be Amerindian, Asian, Black, Hispanic, Mexican, Puerto Rican, White, or other. The UGPA is a continuous variable ranging from 0 to 4. LSAT is an integer-based attribute with a range of $[0, 60]$. FYA, which is the target variable for prediction, is a real number ranging from -4 to 4 (it has been normalized). In this study, we consider $S$ as the sensitive attribute, while $R, G$, and $L$ are treated as features.

**Causal Model**: We adopt the causal model as presented in Kusner et al. (2017). A visual representation of this model can be seen in Figure 4a.

In this causal graph, $K$ represents an unobserved variable, which can be interpreted as *knowledge*. Thus, the model suggests that students' grades (UGPA, LSAT, FYA) are

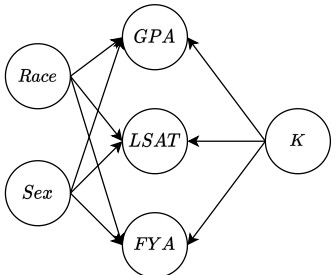

(a) Causal Model for the Law School Dataset

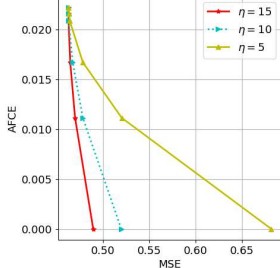

(b) Trade-off between AFCE and MSE

Figure 4: Causal Model and FACE-MSE trade-off for Law School dataset

influenced by their sex, race, and underlying knowledge. We assume that the prior distribution for $K$ follows a normal distribution, denoted as $\mathcal{N}(0, 1)$. The structural equations governing the relationships are given by:[5]

$$
\begin{aligned}
G &= \mathcal{N}(w_G^K K + w_G^R R + w_G^S S + b_G, \sigma_G), \\
L &= Poisson(\exp\{w_L^K K + w_L^R R + w_L^S S + b_L\}), \\
F &= \mathcal{N}(w_F^K K + w_F^R R + w_F^S S, 1).
\end{aligned}
\tag{13}
$$

**Implementation**: Note that race is an immutable characteristic. Therefore, we assume that the individuals only adjust their knowledge $K$ in response to the prediction model $\hat{Y}$. That is $K' = K + \eta \nabla_K \hat{Y}$. In contrast to synthetic data, the parameters of structural equations are unknown, and we have to use the training dataset to estimate them. Following the approach of Kusner et al. (2017), we assume that $G$ and $F$ adhere to Gaussian distributions centered at $w_G^K K + w_G^R R + w_G^S S + b_G$ and $w_F^K K + w_F^R R + w_F^S S$, respectively. Note that $L$ is an integer, and it follows a Poisson distribution with the parameter $\exp\{w_L^K K + w_L^R R + w_L^S S + b_L\}$. Using the Markov chain Monte Carlo (MCMC) method Geyer (1992), we can estimate the parameters and the conditional distribution of $K$ given $(R, S, G, L)$. For each given data, we sampled $m = 500$ different $k$'s from this conditional distribution. We partitioned the data into training, validation, and test sets with $60\%/20\%/20\%$ ratio.

**Results**: Table 2 illustrates the results with $\eta = 10$ and $p_1 = \frac{T}{4}$ and $p_1 = \frac{T}{2}$. In this experiment, $T$ is equal to $\frac{1}{(w_K^F)^2}$. Based on this table, our method achieves a similar MSE as the CF predictor. However, it can improve AFCE significantly compared to the baselines. Figure5 shows the distribution of $Y$ and $Y'$ for the first data point in the test set in the factual and counterfactual worlds. Under the UF and CF predictor, the disparity between factual and factual $Y'$ remains similar to the disparity between factual and counterfactual $Y$. On the other hand, the disparity between factual and counterfactual $Y'$ under our algorithms gets better for both $p_1 = T/2$ and $p_1 = T/4$. Lastly, Figure 4b demonstrates that for the law school dataset, the trade-off between MSE and AFCE can be adjusted by changing hyperparameter $p_1$.

Table 2: Results on Law School Dataset: comparison with two baselines, unfair predictor (UF) and counterfactual fair predictor (CF), in terms of performance (MSE) and LCF (AFCE ,UIR).

| Method | MSE | AFCE | UIR |
|---|---|---|---|
| UF | $0.393 \pm 0.046$ | $0.026 \pm 0.003$ | $0\% \pm 0$ |
| CF | $0.496 \pm 0.051$ | $0.026 \pm 0.003$ | $0\% \pm 0$ |
| Ours ($p_1 = T/4$) | $0.493 \pm 0.049$ | $0.013 \pm 0.002$ | $50\% \pm 0$ |
| Ours ($p_1 = T/2$) | $0.529 \pm 0.049$ | $0.000 \pm 0.000$ | $100\% \pm 0$ |

## 5 CONCLUSION

In this work, we studied the impact of machine learning decisions on individuals' future status using a counterfactual inference framework. In particular, we observed that imposing the original counterfactual fairness may not decrease the disparity with respect to individuals' future status. As a result, We introduced the lookahead counterfactual fairness (LCF) notion, which takes into account the downstream effects of ML models and requires the individual future status to be counterfactually fair. We proposed an algorithm to train an ML model under LCF and studied the impact of such an ML model on individuals's future outcomes through extensive empirical study.

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

## A APPENDIX

### A.1 PROOF OF THEOREM 3.1 AND THEMOREM 3.2

*Proof.* For any given $x, a$, we can find the conditional distribution $U_X | X = x, A = a$ and $U_Y | X = x, A = a$ based on causal model $\mathcal{M}$. Consider sample $u = [u_X, u_Y]$ drawn from this conditional distribution. For this sample, we have,

$$\check{x} = \alpha \odot u_X + \beta \check{a}$$

$$\check{y} = w^\mathrm{T} \check{x} + \gamma u_Y$$

So, the gradient of $g(\check{y}, u_X, u_Y)$ w.r.t. $u_X, u_Y$ are

$$\nabla_{u_X} g = \frac{\partial g(\check{y}, u_X, u_Y)}{\partial u_X} + \frac{\partial g(\check{y}, u_X, u_Y)}{\partial \check{y}} \odot w \odot \alpha \tag{14}$$

$$\nabla_{u_Y} g = \frac{\partial g(\check{y}, u_X, u_Y)}{\partial u_Y} + \frac{\partial g(\check{y}, u_X, u_Y)}{\partial \check{y}} \gamma. \tag{15}$$

Then, $y'$ can be calculated using response function $r$ as follows,

$$y' = y + \eta w^\mathrm{T} (\alpha \odot \frac{\partial g(\check{y}, u_X, u_Y)}{\partial u_X}) + \eta ||w \odot \alpha||^2 \frac{\partial g(\check{y}, u_X, u_Y)}{\partial \check{y}} +$$
$$\eta \gamma \frac{\partial g(\check{y}, u_X, u_Y)}{\partial u_Y} + \eta \gamma^2 \frac{\partial g(\check{y}, u_X, u_Y)}{\partial \check{y}}. \tag{16}$$

Similarly, we can calculate counterfactual value $\check{y}'$ as follows,

$$\check{y}' = \check{y} + \eta w^\mathrm{T} (\alpha \odot \frac{\partial g(y, u_X, u_Y)}{\partial u_X}) + \eta ||w \odot \alpha||^2 \frac{\partial g(y, u_X, u_Y)}{\partial y} +$$
$$\eta \gamma \frac{\partial g(y, u_X, u_Y)}{\partial u_Y} + \eta \gamma^2 \frac{\partial g(y, u_X, u_Y)}{\partial y} \tag{17}$$

Note that the following hold for $g$,

$$\frac{\partial g(\check{y}, u_X, u_Y)}{\partial u_X} = \frac{\partial g(y, u_X, u_Y)}{\partial u_X} \tag{18}$$

$$\frac{\partial g(\check{y}, u_X, u_Y)}{\partial u_Y} = \frac{\partial g(y, u_X, u_Y)}{\partial u_Y} \tag{19}$$

Thus,

$$|\check{y}' - y'| = |\check{y} - y + \eta(||w \odot \alpha||^2 + \gamma^2)(\frac{\partial g(y, u_X, u_Y)}{\partial y} - \frac{\partial g(\check{y}, u_X, u_Y)}{\partial \check{y}})| \tag{20}$$

Given above equation, now we can prove Theorem 3.1 and Corollary 3.2,

- For $g$ in Theorem 3.1, we have,

$$g(\check{y}, u_X, u_Y) = p_1 \check{y}^2 + p_2 \check{y} + p_3 + h(u) \tag{21}$$

$$\frac{\partial g(\check{y}, u_X, \bar{u}_Y)}{\partial \check{y}} = 2 p_1 \check{y}. \tag{22}$$

Equations 20 and 22 together imply that,

$$|y' - \check{y}'| = |y - \check{y} + \check{y} - y| = 0 \tag{23}$$

Since, for any realization of $u$, the above equation holds, we can conclude that the following holds,

$$\Pr(\hat{Y}_{A \leftarrow a}(U) = y | X = x, A = a) = \Pr(\hat{Y}_{A \leftarrow \check{a}}(U) = y | X = x, A = a) \tag{24}$$

- For $g$ in Theorem 3.2, since $g(\check{y}, u_x, u_y)$ is strictly convex in $\check{y}$, we have,

$$(\check{y} - y)(\frac{\partial g(y, u_X, u_Y)}{\partial y} - \frac{\partial g(\check{y}, u_X, u_Y)}{\partial y}) < 0 \tag{25}$$

Note that derivative of $g(\check{y}, u_x, u_y)$ with respect to $\check{y}$ is $K$-Lipschitz continuous in $\check{y}$,

$$|\frac{\partial g(y, u_X, u_Y)}{\partial y} - \frac{\partial g(\check{y}, u_X, u_Y)}{\partial \check{y}}| < \frac{2|y - \check{y}|}{\eta(||w \odot \alpha||^2 + \gamma^2)} \tag{26}$$

we proved that

$$|y' - \check{y}'| < |y - \check{y}| \tag{27}$$

So we have

$$\Pr(\{|Y'_{A \leftarrow a}(U) - Y'_{A \leftarrow \check{a}}(U)| < |Y_{\leftarrow \check{a}}(U) - Y_{\leftarrow \check{a}}(U)|\}|X = x, A = a) = 1 \tag{28}$$

$\square$

## A.2 THEOREM 3.2 FOR NON-BINARY $A$

Let $\{a\} \cup \{\check{a}^{[1]}, \check{a}^{[2]}, ..., \check{a}^{[m]}\}$ be a set of all possible values for $A$. Let $\check{Y}^{[j]}$ be the counterfactual random variable associated with $\check{a}^{[j]}$ given observation $X = x$ and $A = a$. Then, $g(\frac{\check{Y}^{[1]} + ... \check{Y}^{[m]}}{m}, U)$ satisfies LCF, where $g$ defined in Theorem 3.2.

*Proof.* For any given $x, a$, we assume the set of counterfactual $a$ is $\{\check{a}^{[1]}, \check{a}^{[2]}, ..., \check{a}^m\}$. Consider sample $u = [u_X, u_Y]$ drawn from the condition distribution of $U_X|X = x, A = a$ and $U_Y|X = x, A = a$, with a predictor $g(\frac{\check{y}^{[1]} + ... \check{y}^{[m]}}{m}, u)$, use the same way in A.1, we can get

$$|\check{y}'^{[j]} - y'| = |\check{y}^{[j]} - y + \eta(||w \odot \alpha||^2 + \gamma^2)(\frac{\partial g(\check{y}^{[1]} + \cdots \check{y}^{[m]}, u)}{\partial \check{y}^{[1]} + \cdots \check{y}^{[m]}} - \tag{29}$$

$$\frac{\partial g(y + \check{y}^{[1]} + \cdots + \check{y}^{[j-1]} + \check{y}^{[j+1]} \cdots \check{y}^{[m]}, u)}{\partial y + \check{y}^{[1]} + \cdots \check{y}^{[j-1]} + \check{y}^{[j+1]} \cdots \check{y}^{[m]}})| \tag{30}$$

When $y > \check{y}^{[j]}$, we have

$$\check{y}^{[1]} + \cdots \check{y}^{[m]} < y + \check{y}^{[1]} + \cdots \check{y}^{[j-1]} + \check{y}^{[j+1]} \cdots \check{y}^{[m]} \tag{31}$$

and when $y < \check{y}^{[j]}$,

$$\check{y}^{[1]} + \cdots \check{y}^{[m]} > y + \check{y}^{[1]} + \cdots \check{y}^{[j-1]} + \check{y}^{[j+1]} \cdots \check{y}^{[m]} \tag{32}$$

Because $g$ is strictly convex and Lipschitz continuous, we have

$$|\check{y}'^{[j]} - y'| < |\check{y}^{[j]} - y| \tag{33}$$

So we proved that, for any $j \in \{1, 2, ..., m\}$

$$\Pr(\{|Y'_{A \leftarrow a}(U) - Y'_{A \leftarrow \check{a}^{[j]}}(U)| < |Y_{\leftarrow \check{a}}(U) - Y_{\leftarrow \check{a}^{[j]}}(U)|\}|X = x, A = a) = 1 \tag{34}$$

$\square$

## A.3 FORMAL VERSION OF THEOREM 3.3

Consider a non-linear causal model $\mathcal{M} = (U, V, F)$, where $U = \{U_{X, U_Y}\}$, $U_X = [U_1, U_2, ..., U_d]^T, V = \{A, X, Y\}, X = [X_1, X_2, ..., X_d]^T$, $A \in \{a_1, a_2\}$ is a binary sensitive attribute. Assumed that the structural functions are given by,

$$X = A(\alpha \odot U_X + \beta) \quad Y = w^T X + \gamma U_Y \tag{35}$$

where $\alpha = [\alpha_1, \alpha_2, ..., \alpha_d]^T$, $\beta = [\beta_1, \beta_2, ..., \beta_d]^T$, and $\odot$ denotes the element wise production. A predictor $g(\check{Y})$ leads to future outcome $Y'$ that satisfy constraints 7 if $g$ and the causal model has the following three properties

- The value domain of $A$ satisfies $a_1 a_2 \geq 0$.
- $g(\check{y})$ is strictly convex.
- The derivate of $g(\check{y})$ is $K$-Lipschitz continuous with $K \leq \frac{2}{\eta(a_1 a_2 ||w \odot \alpha||^2 + \gamma^2)}$, $|\frac{\partial g(\check{y}_1)}{\partial \check{y}_2} - \frac{\partial g(\check{y}_2)}{\partial \check{y}_2}| < K|\check{y}_1 - \check{y}_2|$.

## A.4 PROOF OF THEOREM 3.3

*Proof.* From the causal functions defined in Section A.3, given any $x, a$, we can find the conditional distribution $U_X|X = x, A = a$ and $U_Y|X = x, A = a$. Similar to the proof of Theorem 3.2, we have

$$\check{x} = \check{a}(\alpha \odot u_X + \beta) \tag{36}$$

$$\check{y} = w^{\mathrm{T}}\check{x} + \gamma u_Y \tag{37}$$

So, the gradient of $g(\check{y})$ w.r.t $u_X, u_Y$ are

$$\nabla_{u_X} g = \frac{\partial g(\check{y})}{\partial \check{y}} \check{a} w \odot \alpha \tag{38}$$

$$\nabla_{u_Y} g = \frac{\partial g(\check{y})}{\partial \check{y}} \gamma \tag{39}$$

Then, $y'$ can be calculated using the response function $r$ as follows,

$$y' = y + \eta(a\check{a}||w \odot \alpha||^2 + \gamma^2)\frac{\partial g(\check{y})}{\partial \check{y}} \tag{40}$$

In the counterfactual world,

$$\check{y}' = \check{y} + \eta(\check{a}a||w \odot \alpha||^2 + \gamma^2)\frac{\partial g(y)}{\partial y} \tag{41}$$

So,

$$|y' - \check{y}'| = |y - \check{y} + \eta(a\check{a}||w \odot \alpha||^2 + \gamma^2)(\frac{\partial g(\check{y})}{\partial \check{y}} - \frac{\partial g(y)}{\partial y})| \tag{42}$$

Because $A$ is a binary attributes, we have

$$a\check{a} = a_1 a_2 \tag{43}$$

From the property of $g$, we have

$$(y - \check{y})(\frac{\partial g(\check{y})}{\partial \check{y}} - \frac{\partial g(y)}{\partial y}) < 0 \tag{44}$$

Note that the derivate of $g(\check{y})$ is $K$-Lipschitz continuous,

$$|\frac{\partial g(\check{y})}{\partial \check{y}} - \frac{\partial g(y)}{\partial y}| < \frac{2|\check{y} - y|}{\eta(a\check{a}||w \odot \alpha||^2 + \gamma^2)} \tag{45}$$

which is to say, for every $u$ sampled from the conditional distribution, $|\check{y}' - y'| < |\check{y} - y|$. So we proved

$$\Pr(\{|Y'_{A\leftarrow a}(U) - Y'_{A\leftarrow \check{a}^{[j]}}(U)| < |Y_{\leftarrow \check{a}}(U) - Y_{\leftarrow \check{a}^{[j]}}(U)|\}|X = x, A = a) = 1 \tag{46}$$

$\square$

## A.5 PROOF ABOUT CF NOT GUARANTEE LCF

**Theorem A.1.** Consider a structural causal model $\mathcal{M}(U, V, F)$ and a response function $r$ with which

$$U' = r(U, \hat{Y})$$

$$X'_r = r(X_r, \hat{Y}), \quad X_r \subset V \text{ are the root nodes}$$

We assume the causal equation determine the target attribute $Y$ is $f_Y$ could be wrriten in the form of $f_Y^r$, where the inputs attributes are all root nodes, i.e.

$$Y = f_Y^r(U, X_r, A) \tag{47}$$

[6] If $\mathcal{M}$ satisfies

$$\Pr(Y_{A \leftarrow a}(U) = y | X = x, A = a) \neq \Pr(Y_{A \leftarrow a'}(U) = y | X = x, A = a) \tag{48}$$

and the response function $r$ is only depend on the value of its inputs, then we have LCF would be violated with a CF predictor, i.e.

$$\Pr(Y'_{A \leftarrow a}(U) | X = x, A = a) \neq \Pr(Y'_{A \leftarrow \check{a}}(U) = y | X = x, A = a) \tag{49}$$

*Proof.* Suppose the conditional distribution of $U$ could be simply denoted as $P_c(U)$, we have

$$\Pr(Y_{A \leftarrow a}(U) | X = x, A = a) = \sum_{u \in \{u | f(u, x_r, a)\} = y} P_c(u) \tag{50}$$

and

$$\sum_{u \in \{u | f(u, x_r, a)\} = y} P_c(u) \neq \sum_{u \in \{u | f(u, x_r, a')\} = y} P_c(u) \tag{51}$$

Because the predictor satisfies CF,

$$U'_{A \leftarrow a} = r(U, \hat{Y}) \tag{52}$$

$$U'_{A \leftarrow \check{a}} = r(U, \hat{Y}) \tag{53}$$

The future outcome could be written as

$$\Pr(Y'_{A \leftarrow a}(U) | X = x, A = a) = \sum_{u | \{f(r(u, \hat{y}), r(x_r, \hat{y}), a) = y\}} P_c(U) \tag{54}$$

From Eq.51, we have

$$\sum_{u | \{f(r(u, \hat{y}), r(x_r, \hat{y}), a) = y\}} P_c(U) = \neq \sum_{u | \{f(r(u, \hat{y}), r(x_r, \hat{y}), a') = y\}} P_c(U) \tag{55}$$

which is to say

$$\Pr(Y'_{A \leftarrow a}(U) | X = x, A = a) \neq \Pr(Y'_{A \leftarrow \check{a}}(U) = y | X = x, A = a) \tag{56}$$

$\square$

## A.6 THEOREM FOR ANOTHER FAMILY OF NON-LINEAR CAUSAL MODEL

**Theorem A.2.** Consider a causal model $\mathcal{M}(U, V, F)$, where $U = (U_X, U_Y)$, $V = \{A, X, Y\}$. Assumed that the structural functions are given by,

$$X = f(\alpha U_X + \beta A) \quad Y = wX + \gamma U_Y$$

in which $f$ is a non-linear function. Then the predictor

$$\hat{Y} = \lambda_1 \check{Y} + \lambda_2 g(U_X, U_Y)$$

, in which $g$ is an arbitary function, improves lookahead counterfactual fairness if $f$ satisfies

- $f$ is strictly convex

- for any $s_1, s_2, s'_1, s'_2$, if $|s_1 - s_2| < |s'_1 - s'_2|$, we have $|f(s_1) - f(s_2)| < |f(s'_1) - f(s'_2)|$

---

[6]The function does not mean there must be a direct effect from $A$ to $Y$. For example, $Y = k_1 U + k_2 X_r + k_3 A$ and $k_3 = 0$

- $f'$ is $K$-Lipschitz continuous with $K < \frac{2}{\lambda_1 w \eta \alpha^2}$

*Proof.* For any given $x, a$, we can find the conditional distribution $U_X | X = x, A = a$ and $U_Y | X = x, A = a$ based on causal model $\mathcal{M}$. Consider a sample $u = [u_X, u_Y]$ drawn from this conditional distribution. For this sample, we have

$$\check{x} = f(\alpha u_X + \beta a)$$

$$\check{y} = w\check{x} + \gamma u_Y$$

So, the gradient of $g(\check{y}, u_X, u_Y)$ w.r.t. $u_X, u_Y$ are

$$\nabla_{u_X} g = \lambda_1 w a f'(\alpha u_X + \beta \check{a}) + \lambda_2 \frac{\partial g(u_X, u_Y)}{\partial u_X}$$

$$\nabla_{u_Y} g = \lambda_1 \gamma + \lambda_2 \frac{\partial g(u_X, u_Y)}{\partial u_Y}$$

Then $y'$ can be calculated using response function $r$ as follows,

$$\begin{aligned}
y' = & w f(\alpha u_X + \lambda_1 w \eta \alpha^2 f'(\alpha u_X + \beta \check{a}) + \lambda_2 \alpha \frac{\partial g(u_X, u_Y)}{\partial u_X} + \beta a) \\
& + \gamma(u_Y + \lambda_1 \eta \gamma + \lambda_2 \eta \frac{\partial g(u_X, u_Y)}{\partial u_Y})
\end{aligned}$$

Similarly, we can calculate counterfactual value $\check{y}'$ as follows,

$$\begin{aligned}
\check{y}' = & w f(\alpha u_X + \lambda_1 w \eta \alpha^2 f'(\alpha u_X + \beta a) + \lambda_2 \alpha \frac{\partial g(u_X, u_Y)}{\partial u_X} + \beta \check{a}) \\
& + \gamma(u_Y + \lambda_1 \eta \gamma + \lambda_2 \eta \frac{\partial g(u_X, u_Y)}{\partial u_Y})
\end{aligned}$$

So,

$$\begin{aligned}
|y' - \check{y}'| = & |w| |f(\alpha u_X + \lambda_1 w \eta \alpha^2 f'(\alpha u_X + \beta \check{a}) + \lambda_2 \alpha \frac{\partial g(u_X, u_Y)}{\partial u_X} + \beta a) \\
& - f(\alpha u_X + \lambda_1 w \eta \alpha^2 f'(\alpha u_X + \beta a) + \lambda_2 \alpha \frac{\partial g(u_X, u_Y)}{\partial u_X} + \beta \check{a})|
\end{aligned}$$

Since $f$ is strictly convex,

$$[(\alpha u_X + \beta a) - (\alpha u_X - \beta \check{a})] \cdot [f'(\alpha u_X + \beta a) - f'(\alpha u_X - \beta \check{a})] < 0$$

And because $f'$ is strictly continuous,

$$\begin{aligned}
(\alpha u_X + \lambda_1 w \eta \alpha^2 f'(\alpha u_X + \beta \check{a}) + \lambda_2 \alpha \frac{\partial g(u_X, u_Y)}{\partial u_X} + \beta a) - \\
(\alpha u_X + \lambda_1 w \eta \alpha^2 f'(\alpha u_X + \beta a) + \lambda_2 \alpha \frac{\partial g(u_X, u_Y)}{\partial u_X} + \beta \check{a}) < \\
(\alpha u_X + \beta a) - (\alpha u_X - \beta \check{a})
\end{aligned}$$

From the second property of $f$, we know that

$$|y' - \check{y}'| < |w f(\alpha u_X + \beta a) - w f(\alpha u_X - \beta \check{a})|$$

which is exactly $|y - \check{y}|$. $\qquad\square$

## B    LEARNING UNDER LCF WITHOUT KNOWN EXOGENOUS VARIABLES

In many real world applications, exogenous variables are remains unknown, where the causal model is unidentifiable in general. Nasr-Esfahany et al. (2023) proposed a kind of model called Bijective Causal Model (BCM). Every structural function

$$V_i = f_i(pa(V_i), U_i)$$

is a bijective function of $U_i$ with a realization of $pa(V_i)$. Even when the prior distribution of $U_i$ remains unknown, with some assumptions, Nasr-Esfahany et al. (2023) was able to infer the counterfactual $\check{V}_i$.

With a SCM defined in Theorem 3.1, we assume that the structural functions are bijective. Suppose the $\check{X}$ and $\check{Y}$ are the counterfactual quantities inferred from Nasr-Esfahany et al. (2023), a predictor

$$g(\check{Y}, \check{X}, X) = p_1\check{Y}^2 + p_2\check{Y} + p_3\check{Y} + h(s(X, \check{X}))$$

in which $s$ is a symmetric function and $h$ is an arbitary function, satifies LCF in Definition 2.1. Since $f_i$ is an invertiable function, we no longer need to infer the exogenous variables $U_i$. The relationship between $\check{X}$ and $X$ can be learned from the observational data as $f_i(pa(X), f_i^{-1}(X))$.

## C    PARAMETERS FOR SYNTHETIC DATA SIMULATION

When generating the synthetic data, we used $\alpha$ = [0.37454012, 0.95071431, 0.73199394, 0.59865848, 0.15601864, 0.15599452, 0.05808361, 0.86617615, 0.60111501, 0.70807258]$^\mathrm{T}$. $\beta$ =[0.02058449, 0.96990985, 0.83244264, 0.21233911, 0.18182497, 0.18340451, 0.30424224, 0.52475643, 0.43194502, 0.29122914]$^\mathrm{T}$. $w$ =[0.61185289, 0.13949386, 0.29214465, 0.36636184, 0.45606998, 0.78517596, 0.19967378, 0.51423444, 0.59241457, 0.04645041]$^\mathrm{T}$. $\gamma$ = 0.60754485 (These values are generated randomly).

## D    DENSITY PLOT FOR LAW SCHOOL DATA



Figure 5: Density plot for $F'$ and $\check{F}'$ under the distribution of $K$ in law school data

