# OpenReview forum: "Counterfactual Fairness With the Human in the Loop"
_ICLR.cc/2024/Conference — Submitted to ICLR 2024_

### Official Review · Reviewer_HGLj · 2023-10-16

**Soundness:** 2 fair
**Presentation:** 2 fair
**Contribution:** 3 good
**Rating:** 5
**Confidence:** 4

**Summary:**

The authors consider the problem of learning a predictive model under the long-term counterfactual fairness constraint where the features of each individual are affected by the prediction. This problem is very important in fairness community, though few causality-based methods have been developed. The proposed fairness criteria seem sound. However, the proposed learning framework seems a bit weak, mainly due to the strong functional assumptions on the underlying structural causal models.

**Strengths:**

- The authors address a very important problem setting.
- Few causality-based methods have been developed in this problem. The only work I know is [Hu+; AAAI2022], which is already cited in this paper.
- The proposed fairness criteria (Definitions 2.1 & 3.2) seem reasonable.

**Weaknesses:**

(A) Formulation of response function $r$

Whereas [Hu+; AAAI2022] use time series data to infer how the feature attributes dynamically change depending on the prediction, this paper makes the functional assumptions about this change, using the existing strategic classification model in ML community. This raises several questions to me:

- **Violation of the SCM definition**: The authors seem to consider the cases where the exogenous variables $U$ are also affected by prediction $\hat{Y}$ (e.g., real-world experiments in Sec. 4.2). However, this contradicts the definition of structural causal model (SCM). As the authors clearly noted in Section 2, in SCM, unobservable (exogenous) variables "are not caused by any variable in $V$". Since the prediction $\hat{Y}$ is usually modeled as one of the endogeneous variables $V$, this completely confuses me. If the authors consider the reduced form of the SCM, then it might be OK. However, for instance, in Example 2.3, each structural equation in the SCM seems to be given as the structural form (i.e., the inputs contain $V$), despite the fact that $U_1'$ and $U_2'$ are exogenous variables. I am not sure whether such an SCM is properly defined. No remark is given for this.

- **Parameter $\eta$**: I am not sure about strategic classification model $r$ at the beginning of Section 3, but is $\eta$ a hyperparameter? This seems a very important factor because it defines how the feature attributes change. If it can be estimated from the data, how can we achieve it?

- **The motivation of response function $r$**: Are there any other function forms for strategic classification models in ML community? If so, what are the advantages and disadvantages of the assumed models?

(B) The proposed learning framework seems ineffective

The authors make restrictive assumptions about the function form of the structural equations (e.g., the linear SCM for feature attributes $X$). I believe that it is OK because this paper is a pioneer work for a novel and important problem setting. However, despite these assumptions (i.e., Eqs. (5) and (6)), the proposed method only addresses parameter $p_2$ and $p_3$ in Eq. (6) and cannot optimize the second-order coefficient $p_1$, and authors explain that it is a hyperparameter.

- **Unlearnable parameter $p_1$**: The optimal value of $p_1$ is given using parameter $\eta$ in $r$. This goes back to my question in (A): Is $\eta$ a hyperparameter? How is it determined? Can the proposed method optimize both $\eta$ and $p_1$?

- **The form of distribution $P(U)$**: Does the proposed method require the knowledge about $P(U)$?
1. There is no description about this in Theorem 3.1. I am not sure, but is the SCM in Eq. (5) deterministic?
2. I believe that performing MCMC at line 2 in Alg. 1 requires the form of $P(U)$. In practice, how can we obtain it? Although the real-world data experiments in Sec. 4.2 seem to consider the cases where it is available, it is usually unknown, isn't it?

(C) Clarity issues

Authors nicely explain the motivation of this work using several examples. I recommend elaborating these examples:

- Example 2.2 is a very important example for illustrating the importance of long-term fairness. However, it seems a bit unclear.
1. Please clearly illustrate what features $A$ and $X$ can be considered in this example. For instance, is a credict score included in $X$?
2. The sentence connection between "Assume that ..." and "However, in the counterfactual world, ..." is unclear. Why is he/she not qualified? Please elaborate more.

- Example 2.3 is not intuitive. The description in Footnote 2 is correct if the SCM is given. In usual, however, since the SCM is unknown, it is difficult to imagine how we can construct the prediction model only with unobservable inputs $U$.


(D) Other minor comments

- In Section 2.2, "the prediction $\hat{Y}$ in the factual world should be the same as that in the counterfactual world ..." is not exactly correct because the counterfactual fairness is defined via marginalization over $U$ and hence is probabilistically defined, as stated in the first equation in Sec. 2.2.

- Please clearly state that Theorem 3.1 considers the causal graph structure in Figure 1 (b). In particular, there is no definition of $U_1, ..., U_d$. Please clearly claim the assumption that these exogenous variables do not include unobserved confounders, i.e., $X_i \leftarrow U_i \rightarrow X_j$ ($i \neq j$).

- Add "Assume that" or "Suppose that" to the second sentence in Theorem 3.1.

**Questions:**

See all questions in (A) and (B).

---

> ### Author Response · Authors · 2023-11-18
>
> * A1. We want to emphasize that we treat $U’$ as a hidden variable. However, it is not an exogenous variable.  $U’$ is the effect of $U$ and decision $\hat{Y}$. On the other hand, $U$ is an exogenous variable and does not have any parents. In this case, SCM is not violated.
>
> * A.2 $\eta$ is not a hyper-parameter. We consider it as a constant. It is a value that reflects how the individual response to the current prediction. In our paper, we assumed it is a given since we don't have a real dataset about the interaction. If we measure $Y$ and $Y’$ before and after decision $\hat{Y}$, respectively, $\eta$ can be estimated.
>
> * A.3 There are other forms for strategic classification. For example, when a cost function is defined on an attribute $U_{i}$ as $c(u_{i}, u_{i}’)$ (which means how much the individual need to cost by changing the attribute), the response function of this attribute is the $\arg \max_{u_{i'}} \hat{Y}(u_{i}’) - c(u_{i}, u_{i'})$.
>
>    The advantage for the response function form we used in our paper is that it is simple yet can reflect a general kind of people’s response. It means that people try to take action to change their status along the direction that will change the prediction mostly. The disadvantage is that when the prediction model is not differentiable, the response function could be out of work.
>
> * B.1  $p_{1}$ is a hyper-parameter that controls the trade-off between the LCF and accuracy. For example, if $p_{1} = \frac{1}{2\eta(||w\odot \alpha||^{2} + \gamma^{2})}$, LCF (eq.4) is satisfied and we have perfect fairness. If we choose smaller $p_1$, we improve accuracy but the LCF is satisfied approximately (I.e., the left hand side in eq.4 is close to the right hand side).
>
>    On the other hand, $\eta$ is fixed in our model, we cannot and don't need to optimize it.
>
> * B.2   Yes, we need the knowledge about the prior distribution $P(U)$.
>
> * B.3  The Eq.5 is deterministic meaning that $\alpha, \beta, \gamma$ and $w$ are fixed. However, we do not know the specific value of $U$ and it is a random variable.
>
> * B.4 Yes, when we used MCMC to get the conditional distribution of $U$, we need to know $P(U)$. It is a common requirement for counterfactual fairness. For the experiment, we used the same prior as the one used in the experiment of Kunser et al. 2017 .
>
> * C.1. $A$ is the sensitive attribute which determines different demographic groups.  For example, it can be the gender of the person. $X$ are the features, for example, the income and credit score. We are trying to predict whether the person will default or not, and the credit score it is also included in $X$. Note that credit score is not the same as default probability.
>
> * C.2 We can imagine a case, we can observe the income of a person. For simplicity, assume that the income dictates the qualification of an applicant. And there is a hidden attribute U represents how hard the person works. Both $U$ and $A$ can impact income. So, in the counterfactual world where the person has the same $U$ with different gender may have a different income. If that income is low, the counterfactual person would not be qualified.
>
>    Mathematically, $Y_{A\leftarrow a}(U)$ and $Y_{A\leftarrow a’}(U)$ do not have the same distribution necessarily. However, counterfactual fairness implies that the predictions $\hat{Y}_{A\leftarrow a}(U)$ and $\hat{Y}_{A\leftarrow a'}(U)$ should have the same distribution.
>
> * C.3 We assumed that SCM is known in our problem which is a common practice (e.g., Kusner et al. 2017)
> * D.1 Sorry, you are correct. We meant the distribution of \hat{Y}. We have corrected it in the newly updated version.
> * D.2 The assumption is implicitly included in Eq.5. We also state it explicitly in theorem 3.1.
> * D.3 Thank you for your reminding about the clarity. We added it to our updated version.

---

> > ### Comment · Reviewer_HGLj · 2023-11-22
> > **Response**
> >
> > Thank you for your response.
> >
> > Unfortunately, I will reduce my overall rating because the proposed method seems much less practical than I expected.
> >
> > According to the authors' response, the proposed method requires the knowledge about exogenous variables, $P(U)$, which is rarely available in practice. Although the authors claim that it is a common requirement, I believe it is not true. Most causality-based fairness methods do not need this assumption, other than several exceptions of pioneer work like [Kusner+; 2017].
> >
> > As I already pointed out, the functional assumptions of the proposed method seem demanding, and the practical benefit under real-world applications is unclear.

---

> > > ### Author Response · Authors · 2023-11-23
> > > **Regarding $P(U)$**
> > >
> > > We fully understand your concerns regarding the use of $P(U)$. It's important to note that our algorithm does not necessarily require to use $P(U)$. Under certain conditions, counterfactual quantities (e.g., $\check{X}$ and $\check{Y}$) can be effectively calculated without any knowledge about $U$. For instance, if we are working with bijective causal models [1], it is possible to estimate counterfactual $\check{X}$ using only observable variables. We have included a detailed explanation of this adaptation in Appendix B of our revised edition.
> > >
> > > [1] Nasr-Esfahany et al., Counterfactual Identifiability of Bijective Causal Models, ICML 2023.

---

> > > > ### Comment · Reviewer_HGLj · 2023-11-23
> > > > **Not sure**
> > > >
> > > > I know that by making the functional assumptions on the SCMs, we can infer the counterfactual distributions.
> > > >
> > > > However, since the authors have already made the linearity assumptions on the SCMs and the linear functions are invertible (i.e., one-to-one), if the authors apply the inference technique such as [1], without making bijective causal model assumptions, it seems possible to infer the distribution $P(U)$. Am I correct?
> > > >
> > > > If so, the further revision should be made. Anyway, I cannot follow the inconsistent responses by the authors, so I am not sure whether the paper is acceptable.

---

> ### Author Response · Authors · 2023-11-23
> **Regarding $P(U)$**
>
> If you mean the prior distribution of $U$ by $P(U)$, even with the bijective assumption, it is still impossible to infer it without knowing the distribution of all the observable variables.
>
> You might think it is inconsistent because we said we need $P(U)$ at first response but not now. It is because we thought you are talking about a general counterfactual inference. Without bijective assumption, we can use methods like MCMC to infer the counterfactual values ($\check{X}$, $\check{Y}$), which needs the prior distribution $P(U)$. But with bijective assumption, we can apply the method in [1] to estimate $\check{X}$ or $\check{Y}$ without $P(U)$. Then we can use what we described in Appendix B to solve the following problems.
>
> We appreciate your help. Your valuable comments can help us to improve the paper during the discussion period.  If you see any inconsistency, we are going to address it.

---

> > ### Comment · Reviewer_HGLj · 2023-11-23
> > **Response**
> >
> > My notation P(U) might be confusing.
> >
> > Now I understand what you mean, which is what I expected in your first response.
> >
> > I will correct my above question as follows:
> >
> > Since the authors have already made the linearity assumptions on the SCMs and the linear functions are invertible (i.e., one-to-one), if the authors apply the inference technique such as [1], without making additional bijective causal model assumptions, it seems possible to infer counterfactual values. Am I correct?

---

> > > ### Author Response · Authors · 2023-11-23
> > >
> > > Thanks for your response. Yes, you are right. For the linear SCM, we do not need additional assumption to infer counterfactual values.

---

### Official Review · Reviewer_VAjK · 2023-10-28

**Soundness:** 1 poor
**Presentation:** 3 good
**Contribution:** 1 poor
**Rating:** 3
**Confidence:** 4

**Summary:**

This paper revisits counterfactual fairness (CF) by Kusner et al. 2017 by considering the ML model’s impact on the behavior of the individual in the long-term. Under CF, we only consider a static setting and whether the ML model’s predictions $\hat{Y}$ are counterfactually fair. Here, the work considers the case where a counterfactually fair model induces a future outcome $Y’$ (the outcome $Y$ is used for training the model) that is not counterfactually fair. It introduces the fairness notion of lookahead counterfactual fairness (LCF) and proposes an algorithm to insure it for a given ML model. To do so, it assumes strategic agents. The paper tests and compares LCF to CF on two datasets, including Law School Success from Kusner et al. 2017.

Although the topic is relevant, the problem formulation is unclear and (I suspect) at odds with counterfactuals as described by Pearl, meaning non-backtracking or interventional counterfactuals. It needs a stronger and clearer formalization, though I encourage the authors to continue with this line of work. See my comments below: I’ll use S to denote strengths, W to denote weaknesses, and O to denote opportunities for improvements.

I'm willing to increase my score if the authors address my questions/concerns.

**Strengths:**

S1: Evaluating the long-term effects (or, overall, the impact of a deployed model’s prediction) is an extremely important topic. The authors are correct to raise concerns around how having a counterfactually fair model might not mean anything if we don’t consider how individual will reach to that model when deployed.

S2: Similarly, formulating LCF as an in-processing step is interesting as it forces us to train models that are forward looking.

**Weaknesses:**

O1: Based on the title, I thought it would be a work on counterfactual fairness involving a human (i.e.,  AI-Human interactions). It’s unclear to me how the human is in the loop in this work -- also, there’s a typo in the title: Counterfactual Fairness with a Human in the Loop. I suggest changing the title to avoid confusion, maybe something along the lines of “Lookahead Counterfactual Fairness” or “Counterfactual Fairness under Strategic Classification”.

O2: Example 2.1 is welcome, as many causal fairness papers take for granted counterfactual generation, and such a section can help unfamiliar readers. I’d suggest improving it by being explicit about the abduction, action, and prediction steps.

O3: From (1), it seems causal sufficiency is assumed for the paper, no? Please clarify if so, and whether it’s a limitation to the problem formulation. Especially in this setting where the authors are considering time (though this somewhat unclear as well) and the risk of hidden confounders increases.

W1: In the motivating example (Example 2.2: starting from “Assume that this person…”), it’s unclear how there is a counterfactual world not captured by the causal graph used for CF. The counterfactual instance generated there to test for counterfactual fairness is specific to the SCM available or assumed (i.e., the worldview). That same counterfactual should be, in principle, the closest possible world to what is observed, no? How is this individual not qualified in the counterfactual world? Are you implying model bias?

I understand that (correct me if I’m wrong) this individual is unlabeled, but somehow we know that in his/her the factual world $Y=1$ and in the counterfactual world $Y^{CF}=0$, while the model being studied for counterfactual fairness gives $\hat{Y}=\hat{Y}^{CF}=1$. This reasoning I understand, but it’s unclear how and from where this additional information is available. This is important because this ambiguity appears also in the formalization of LFC.

W2: I’m not sure if strategic classification is suitable as a framework for LFC. Under strategic classification, individuals essentially are assumed/expected to cheat: i.e., to lie about a hidden information such that the model changes its decision. Hence, why we train the model to account for such strategic responses. How does this translate into the fairness setting? Are the individuals improving over time or are they in fact just cheating? If so, then the present problem formulation is just strategic classification but with the consideration of counterfactual fairness. Please clarify if I’m wrong.

To me this is clear when looking at the operationalization of LFC, which is based on changing the latent variables $U_i'$ (Section 3: response function for $U_i’$). Under counterfactual fairness, the $U$’s are often interpreted as inherent but unobserved skills to the individual (like aptitude). These are given and materialize through the observed variables (like SAT scores). These variables, though, are exogenous to the system. We can infer them using the abduction step, but how exactly is an individual basing his/her response on a variable that is out of his/her control?

The formulation in Section 3 seems like a backtracking counterfactual formulation where we imagine a setting were the individual can update the exogenous variables (see Sander et al.’s Backtracking Counterfactuals). The issue here then is that counterfactual fairness is built on non-backtracking counterfactuals, meaning once the exogenous variables are set, all interventions occur in the observable space. Therefore, I don’t understand the second term in the response function for $U_i’$.

Further, I’d argue this misconception is due to using the strategic classification framework: these strategic agents are not strategic in that they improve because of $\hat{Y}$ (as in their given aptitude $U_X$ changes and $X$ increases) but are strategic in that they lie about improving so that $\hat{Y}$ changes (the aptitude $U_X$ cannot change, but they can lie about $X$ to give the impression that it does). This type of agent doesn’t seem useful for what LFC aims to achieve.

For example, in the Implementation part for 4.2, the authors argue “we assume that individuals adjust knowledge $K$ in response to the prediction model $\hat{Y}$”. $K$ here is measured through GPA, LSAT, and FYA (as shown in Figure 4(a)) and is exogenously given to the SCM. How exactly can $K$ change under the current SCM?

Further, conceptually, what exactly does it mean for the individual to change $K$ when, according to the SCM, the individual has no other mechanisms to do so? The setup is incomplete here. Maybe, this is where the human-in-the-loop needs to appear, or an enhancement of the causal graph denoting, e.g., an exogenous intervention in the form of an educational program.

W3: Overall, it’s not clear what is the long-term fairness problem LFC addresses (as highlighted by the Law School Success example). What would help LFC is explicitly admitting $\hat{Y}$ into its formulation. The authors claim that the model has a “downstream impact” on the future outcome $Y’$. Doesn't this imply $\hat{Y} \rightarrow Y’$? Where is $\hat{Y}$ in (4)? As it is setup, it states that $X$ and $A$ have (at least) some association to $Y’$ but so does $\hat{Y}$, no? Please clarify.

**Questions:**

See O3, and all Ws.

---

> ### Author Response · Authors · 2023-11-18
>
> ### For question 1:
> Yes, we use human in the loop to emphasize that a human is interacting with an AI model and responds to it through a response function $r$.
>
> We appreciate your opinion about the title. We are willing to change the title to Lookahead Counterfactual Fairness if the ICLR policy allows us to change the title.
>
> ### For question 2:
> Thank you for your comment. We modified the section to clearly indicate the three steps in the updated version.
> Question 3: Yes, causal sufficiency is required. Generally speaking, counterfactual variables are defined based on the complete causal structure. So, we need to measure all the hidden variables. Both counterfactual fairness (Kusner et al. 2017) and lookahead counterfactual fairness can be studied when the causal structure is known. There are various methods for finding a causal model, see e.g., Peters et al. (https://arxiv.org/pdf/1309.6779.pdf).
>
> ### For weakness 1:
> CF considers both factual and counterfactual worlds. However, it imposes constraint on the prediction without considering the consequence of the decision in factual and counterfactual worlds. We are not claiming that counterfactual fairness is a bad fairness measure and has limitations. Just we are saying LCF and CF have different goals.  In particular, we want to emphasize that while \hat{Y} is the same in the factual and counterfactual world under CF constraint, true Y and future Y’ may not be the same in both worlds. If Y is not the same in both worlds, and we want to make sure that Y’ will be the same in the factual and counterfactual world, then CF fairness constraint cannot handle this situation.  You can see more about why CF does not imply LCF in theorem A.1 we added in the updated version.
>
> In our assumption, the individual is labeled (or at least we can approximate the $Y$     with the causal graph). So, when the individual in the factual world and counterfactual world belongs to the different demographic group (which means different value for the node representing this attribute in the causal graph), it is possible to get $Y = 1$ and $Y^{CF} = 0$ under LCF.
>
> ### For weakness 2:
>  As we know, strategic classification did not always mean cheating. There are two kinds of strategic behavior: manipulation and improvement (see in https://openreview.net/forum?id=W98AEKQ38Y). In this paper, we are considering the case for improvement.
> We want to emphasize that $U$ and $U'$ are not the same variables. $U'$ is the hidden variable after the individual receives a decision. On the other hand, U is the exogenous hidden variable (without any parents) before decision \hat{Y}. U and U' are completely separate variables. U’ can be impacted by $U$ and $\hat{Y}$. As you can see in our model, $U' = r (U, \hat{Y}) $.
>
> If you think of $U$ and $U'$ as two separate variables, then it is easy to see that counterfactuals are non-backtracking in this paper. You can interpret U as an inherent hidden variable (similar to what we have seen in literature) and $U'$ as the hidden variable but it is the effect of $U$ and $\hat{Y}$.
>
> Similarly, $Y$ and $Y'$ are two separate variables. Consider a loan approval problem. The qualification of an applicant at the current time is denoted by $Y$. On the other hand, the qualification of an applicant in the future is denoted by $Y'$. $Y'$ will depend on current circumstances (i.e., $X, U$) as well as loan approval/rejection at the current time ($\hat{Y}$). In other words, $\hat{Y} X$, and $U$ are the cause for $Y'$. However, $\hat{Y}$ is not a cause for $Y$ because $Y$ is the qualification before loan decision.
> Given this explanation, I think our experiment in section 4.2 makes sense. We consider two separate variables $K$ and $K'$. In our setting, the decision $\hat{Y}$ cannot change $K$. $K$ does not change over time because $K$ is the knowledge at the current time. $K'$ is a separate variable and its value is different from $K$.
>
> We respectfully ask the reviewer to look at the paper one more time. We strongly believe that if the reviewer reads the paper one more time and considers that $(U, X, Y) $ and $(U', X', Y') $ are two separate variables, the story of our paper would make more sense. You can also see the strategic classification does not cause any issue as $U$ and $U'$ belong to separate time and are different in nature. $U$ is exogenous and no variable is a cause for $U$. While $U'$ is not exogenous and other variables can be a cause for $U'$.

---

> ### Author Response · Authors · 2023-11-18
>
> ### For weakness 3:
>  We want to emphasize that LCF takes into account $\hat{Y}$. As we mentioned earlier, $Y'$ and $Y$ are two separate variables, and $\hat{Y}$ is a cause for $Y'$. Therefore, by imposing a constraint on $Y'$, we are imposing certain constraints on $\hat{Y}$ as well.
>
> The whole point of this paper is to make sure that $\hat{Y}$ is designed in a way that equation 4 is satisfied. Even though $\hat{Y}$ does not appear in equation 4, it is a cause for $Y'$. Therefore, we need to design $\hat{Y}$ very carefully to satisfy equation 4.

---

> > ### Comment · Reviewer_VAjK · 2023-11-22
> > **Response**
> >
> > I appreciate the authors' detailed responses. Unfortunately, the proposed framework and it's application still remain unclear to me. I'm keeping my score.

---

### Official Review · Reviewer_pE5E · 2023-11-01

**Soundness:** 3 good
**Presentation:** 3 good
**Contribution:** 2 fair
**Rating:** 3
**Confidence:** 4

**Summary:**

The paper considers fair prediction tasks where individuals can respond to predictions by changing their features, and extends the definition of counterfactual fairness to apply to the downstream future value of the outcome variable after this responsive feature change.

The paper provides some theory, describes how to train models to satisfy the new fairness criteria, and explores the use of its ideas on real and synthetic data.

**Strengths:**

The practical problem involving strategic responses and (sequential) fairness over time is well motivated, interesting, and potentially impactful.

The paper is fairly thorough and clear given the space constraints.

The relaxation in Definition 3.2 is interesting, and perhaps could be made a bit stronger by requiring the future difference to be smaller by a certain pre-specified fraction (or addition amount)?

**Weaknesses:**

1) It seems there is a strong assumption regarding no direct effects of A on Y, e.g. in the motivating Example 2.3 with Figure 1, and in the structural equations (5) used in Theorem 3.1 and in Algorithm 1. If this is a necessary assumption of the method it would be appear to be a severe limitation for any fairness fairness (apparently there is no "direct" discrimination?), and if it is not necessary then it should be relaxed.

2) While the application problem is interesting, I think the extension of counterfactual fairness is somewhat incremental. For example, why not simply relabel $Y$ and treat $Y'$ as the actual response for the prediction task? The response function (3) and structural equation for $Y$ are fixed. The intermediate prediction of $Y$ is largely irrelevant, inconsequential since the fairness of $Y'$ is the deciding issue.

3) The main text explores the case of linear causal models. The non-linear case would be more interesting for ICLR, and more generally applicable.

**Questions:**

Could the Example 3.2 be used to make a stronger case for why LCF cannot be considered a minor extension of CF (after renaming $Y$ and treating $Y'$ as the main outcome)?

Could the assumption of no direct discrimination be relaxed? How would the algorithms need to be modified?

Could the paper focus more on the non-linear case?

**Details Of Ethics Concerns:**

The method has some limitations which I believe are not adequately discussed (see e.g. the first comment about assumed causal structure in the weaknesses section above). There is a risk of discriminatory applications if the limitations are not handled properly.

---

> ### Author Response · Authors · 2023-11-18
>
> ### For weakness 1 and question 2:
> We do not make any assumption on whether $A$ is a cause for $Y$ or not. As you can see in figure 1a and example 2.3, $A$ is indeed a cause for $Y$.
>
> ### For weakness 2 and question 1:
>  Note that LCF is not a minor extension of CF. We cannot simply rename $Y$ and treat $Y'$ as the new label. $Y'$ is a consequence of the decision $\hat{Y}$ and depends on both $Y$ and $\hat{Y}$. As discussed in our paper, satisfying LCF is much more challenging, and we cannot achieve LCF using the same approach as for CF. In the new version of our paper, we have also included Theorem A.1 in the appendix to demonstrate that a predictor satisfying CF does not necessarily satisfy LCF.
>
> Furthermore, we want to emphasize that imposing a fairness constraint on $\hat{Y}$ does not necessarily mitigate harm if we do not consider the future outcome/consequence of $\hat{Y}$. For example, in a loan approval problem, having the same selection rate among different demographic groups may seem like a plausible fairness constraint. However, it is possible that the majority of people in one demographic group are unable to repay the loan and default. Consequently, having the same selection rate across different demographic groups may adversely affect one demographic group. Therefore, we must consider the consequences of our decision. In our model, $Y'$ represents the future outcome/consequence of the predictor $\hat{Y}$.
>
> ### For weakness 3 and question 3:
> We have already included Theorem 3.3 for the nonlinear causal model in our paper. Additionally, in the new version, we have added Theorem A.2 to demonstrate that our results can be extended to the nonlinear case.

---

### Official Review · Reviewer_ouSR · 2023-11-03

**Soundness:** 3 good
**Presentation:** 3 good
**Contribution:** 2 fair
**Rating:** 5
**Confidence:** 5

**Summary:**

This work considers a new setting by combining counterfactual fairness and strategic prediction. They propose a new fairness notion that considers the changes in features of users after getting feedback from the first stage. Then they show counterfactually fair predictors are not satisfying their proposed notion LCF. Then they show a simple predictor can satisfy LCF using predictions from the last stage and this predictor can be better than counterfactually fair predictors under the strong assumptions of the causal model. In experiment they verified those claims in the restricted settings.

**Strengths:**

-  Proposed a novel notion for counterfactual fairness under the strategic classification setting. This setting can be useful in some real-world applications.
- Under restricted settings (strong assumptions on the causal models), this work provides theoretical results to show counterfactual fairness is not enough for their setting and a simple predictor can surely outperform counterfactually fair predictor.
- They verified their theoretical results with experiments on two datasets.

**Weaknesses:**

- Identification of counterfactuals: The authors may want to make it clear if they only consider deterministic counterfactual (each individual has deterministic noise U) or probablistic counterfactual (each individual has a distribution of U). Note that without strong assumptions, it is often impossible to identify U, especially when U is probablistic. As U is the input of the LCF predictor g, U has to be identifiable from observational data to make the whole proposed framework work in practice.

- Limited settings: The structural causal model is very restricted with linearity, no hidden variables and additive noise. It would be better to provide results similar to Theorem 3.1 with different assumptions on causal model. In addition, It would be better to discuss more generally under what conditions LCF can be violated when CF is satisfied. For example, what family of functions r will lead to this and what conditions the SCM has to meet.

- Limited baselines and related work: The authors only compare their method with the counterfactually fair predictor from Kusner et al and their literature review is not comprehensive. However, there exists recent work in counterfactual fairness. The authors may want to compare with multiple different methods that aim to achieve counterfactual fairness such as [1,2,3].

[1] Wu, Yongkai, Lu Zhang, and Xintao Wu. "Counterfactual fairness: Unidentification, bound and algorithm." In Proceedings of the twenty-eighth international joint conference on Artificial Intelligence. 2019.

[2] Xu, Depeng, Yongkai Wu, Shuhan Yuan, Lu Zhang, and Xintao Wu. "Achieving causal fairness through generative adversarial networks." In Proceedings of the Twenty-Eighth International Joint Conference on Artificial Intelligence. 2019.

[3] Ma, Jing, Ruocheng Guo, Aidong Zhang, and Jundong Li. "Learning for Counterfactual Fairness from Observational Data." In Proceedings of the 29th ACM SIGKDD Conference on Knowledge Discovery and Data Mining, pp. 1620-1630. 2023.

**Questions:**

- The authors might want to discuss why Def 3.2 compares Y' with Y instead of \hat{Y}, given the counterfactual outcomes Y'_{A\leftarrow a} and Y_{A\leftarrow a} are not identifiable in general unless strong assumptions are. If there is a \hat{Y} in the first step, what is the assumption about the relationship between \hat{Y} and Y and why is there no \hat{Y'} in the second step?

- In Eq.(2), it is clear that P(U|O=o) is inferred from the observed data. However, the paragraph above it says replacing noise U with u, which means intervention on noise U, how is this possible?

---

> ### Author Response · Authors · 2023-11-18
>
> ### For weakness 1:
>  We are considering the probabilistic counterfactual. In our model, we can only infer the distribution of $U$ conditioned on the observed features. We utilized $U$ as the input for our LCF predictor, similar to Kusner et al. 2017 (Section 4.1), for training a counterfactually fair predictor. We sample the value of $U$ from the conditional distribution and take it as the input. Under the distribution of $U$, our algorithm minimizes the expected error over $U$, $X$, and $Y$. For inference, we can also find the expectation of the output over $U$ (similar to Kusner et al. 2017).
>
> ### For weakness 2:
> We aim to emphasize that our method can be extended to nonlinear causal models. In particular, we consider a nonlinear causal model in Theorem 3.3. Additionally, in Appendices A.3 and A.4, we provide a full illustration and proof of the theorem. We also introduce another type of nonlinear causal model in Theorem A.2. In Appendix A.6, we present the formal theorem and a comprehensive proof.
>
> In all the causal models discussed in our paper, we have included hidden variables. For example, in our linear model, we incorporate $U_X$ and $U_Y$ as hidden variables that affect the features, as shown in Eq. 5. These are expressed in the form of additive noise.
>
> In general, CF does not imply LCF. We added theorem A.1 in Appendix A.6 showing that when LCF could be violated even though CF was satisfied.
>
> ### For weakness 3:
> We implemented the method used in Jing et al., and the results are as follows:
>
> |Method|MSE|AFCE|UIR|
> |---|---|---|---|
> |Jing et al.|0.0036 $\pm$ 0.003|1.296 $\pm$ 0.000| 0% $\pm$ 0.000|
>
> We reported the MSE, which reflects the distance between $\hat{Y}$ and $Y$, and the AFCE, which reflects the average distance between $Y'$ and $\check{Y}'$. Compared to our methods, this method achieves slightly better accuracy, but like other counterfactual fair predictors, it cannot improve LCF at all. UIR represents the ratio by which LCF has been improved. It is 0% for this method, while we can achieve 100%.
>
> Note that the method proposed in Depeng et al. is only applicable to a classification setting, whereas in our experiment, we are working with a regression problem. We also want to emphasize that the method proposed in Yongkai et al. is for synthetic data generation under intervention, not for training a predictor. Moreover, we could not find the GitHub repository for the Yongkai et al. paper, and their experiments are not reproducible.
>
> ### For question 1:
> Generally speaking, counterfactual variables are defined based on a causal structure. Both counterfactual fairness (Kusner et al. 2017) and lookahead counterfactual fairness can be studied when the causal structure is known.  There are various method for finding a causal model, see e.g., Peters et al. (https://arxiv.org/pdf/1309.6779.pdf).
>
> There is a $\hat{Y}$ in the first step. Our goal is to train $\hat{Y}$ which can estimate $Y$ under LCF. There is no pre-defined relationship between $\hat{Y}$ and $Y$, but we try to make $\hat{Y}$ and $Y$ as close as possible.
>
> We used $Y'$ in Definition 3.2 instead of $\hat{Y}$ because we want to make the future outcome $Y'$ fair. Imposing fairness constraints on $\hat{Y}$ has been studied in the literature. However, in this work, we aim to take into account the future impact of our decision. For example, consider a loan approval problem. Having the same selection rate among different demographic groups may be a plausible fairness constraint. However, it is possible that the majority of people in one demographic group are not able to repay the loan and default. As a result, having the same selection rate across different demographic groups may hurt one demographic group, and we have to take into account the consequences of our decision. In our model, $Y'$ represents the future outcome/consequence of the predictor $\hat{Y}$.
>
> ### For question 2:
> We are not doing intervention on the noise $U$. We are using the law of total probability with respect to U to find the probability distribution for counterfactual random variable $Y_{Z\leftarrow z}(U)$.

---

> > ### Comment · Reviewer_ouSR · 2023-11-22
> > **For weakness 1**
> >
> > Thanks for the reply.
> >
> > Do the authors mean their method is similar to Kusner et al., which has to assume the causal relations between U and observed variables and has to learn U from data? In more recent literature of counterfactual inference [1], the inference of U can be avoided under certain assumptions, which makes it more practical. The authors may want to improve their method to avoid inferring U which is a challenging task.
> >
> > [1] Nasr-Esfahany, Arash, Mohammad Alizadeh, and Devavrat Shah. "Counterfactual identifiability of bijective causal models." arXiv preprint arXiv:2302.02228 (2023).

---

> > > ### Author Response · Authors · 2023-11-22
> > > **Regarding bijective causal models**
> > >
> > > Thank you for your comment.
> > >
> > > The bijective causal model assumes that there is a one-to-one mapping between V_i (endogenous variable) and U_i (exogenous variable).
> > >
> > > Our method is general. It can be applied to the bijective causal model as well. Do you want us to provide an experiment with a bijective causal model? Does this help you to increase your score?

---

> > > > ### Comment · Reviewer_ouSR · 2023-11-22
> > > >
> > > > I think it is not just experiments, as authors' proposed method need U as the input of the function g, there must be an update on the methodology itself to avoid using U as the input of the model as it is an unobserved variable.
> > > >
> > > > While with bijective models [1], one can infer counterfactual outcome's distributions as they are identified as conditional distributions, where the inference of U is no longer needed.
> > > >
> > > > [1] Nasr-Esfahany, Arash, Mohammad Alizadeh, and Devavrat Shah. "Counterfactual identifiability of bijective causal models." arXiv preprint arXiv:2302.02228 (2023).

---

> > > > > ### Author Response · Authors · 2023-11-22
> > > > > **Regarding using U**
> > > > >
> > > > > We totally agree with you. The Bijective Causal Model can completely address your concern. If we assume that we have a bijective model, we can avoid using U because there is a one-to-one mapping between endogenous variable and exogenous variable.
> > > > >
> > > > > We can add one section to the paper regarding how to avoid using U as the input of our model. Does this address your concern?

---

> > > > > > ### Comment · Reviewer_ouSR · 2023-11-22
> > > > > >
> > > > > > I will consider to increase my score if the authors can provide such a revision. Thanks for the reply.

---

> > > > > > > ### Author Response · Authors · 2023-11-23
> > > > > > >
> > > > > > > We appreciate your comment about inferring counterfactual quantities without using $U$. Assuming the underlying causal model is a Bijective Causal Model (BGM), where the structural functions $f_i(pa(V_i), U_i)$ are bijective, we can derive the counterfactual attribute using $f_i^{-1}$ and $V$ which could be using as the input of our predictor. This allows the predictor to satisfy LCF criteria without relying on $U$. Additional details of this modification are included in Appendix B of our revised paper.

---

### Author Response · Authors · 2023-11-21

Thank you for your comments. We appreciate if you can go over our response and let us know if you have any other questions. We have already revised our manuscripts. The texts with blue color have been recently added to the manuscripts.

---

### Meta-Review · Area_Chair_8613 · 2023-12-12

**Metareview:**

This paper looks at counterfactual fairness in an arguably more realistic setting that addresses the underlying dynamics of the world.  It introduces lookahead counterfactual fairness (LCF) as a method for taking into account an ML model's downstream impacts on the world/stakeholders, and then holding counterfactual fairness as a constraint in that downstream world.  Reviewers all appreciated the problem at hand and the general approach to solving it, but - including after the rebuttal - either maintained or doubled down on concerns of readability and especially practical applicability of the work.  Reviewers and this AC see this work as promising but not presentable within the ICLR timeline.

**Justification For Why Not Higher Score:**

Reviewers maintained their scores after the rebuttal, and those low scores hinged on issues that can't be changed in a non-shepherded rebuttal process.

**Justification For Why Not Lower Score:**

N/A

---

### Decision · Program_Chairs · 2024-01-16

Reject